# WHAT TO PRUNE AND WHAT NOT TO PRUNE AT INITIALIZATION

## ABSTRACT

Post-training dropout based approaches achieve high sparsity and are well established means of deciphering problems relating to computational cost and overfitting in Neural Network architectures citesrivastava2014dropout, (Pan et al., 2016), Zhu & Gupta (2017), LeCun et al. (1990). Contrastingly, pruning at initialization is still far behind Frankle et al. (2020). Initialization pruning is more efficacious when it comes to scaling computation cost of the network. Furthermore, it handles overfitting just as well as post training dropout. It is also averse to retraining losses.

In approbation of the above reasons, the paper presents two approaches to prune at initialization. The goal is to achieve higher sparsity while preserving performance. 1) K-starts, begins with k random p-sparse matrices at initialization. In the first couple of epochs the network then determines the "fittest" of these p-sparse matrices in an attempt to find the "lottery ticket" Frankle & Carbin (2018) p-sparse network. The approach is adopted from how evolutionary algorithms find the best individual. Depending on the Neural Network architecture, fitness criteria can be based on magnitude of network weights, magnitude of gradient accumulation over an epoch or a combination of both. 2) Dissipating gradients approach, aims at eliminating weights that remain within a fraction of their initial value during the first couple of epochs. Removing weights in this manner despite their magnitude best preserves performance of the network. Contrarily, the approach also takes the most epochs to achieve higher sparsity. 3) Combination of dissipating gradients and kstarts outperforms either methods and random dropout consistently.

The benefits of using the provided pertaining approaches are: 1) They do not require specific knowledge of the classification task, fixing of dropout threshold or regularization parameters 2) Retraining of the model is neither necessary nor affects the performance of the p-sparse network.

We evaluate the efficacy of the said methods on Autoencoders and Fully Connected Multilayered Perceptrons. The datasets used are MNIST and Fashion MNIST.

## 1 INTRODUCTION

Computational complexity and overfitting in neural networks is a well established problem Frankle & Carbin (2018), Han et al. (2015), LeCun et al. (1990), Denil et al. (2013). We utilize pruning approaches for the following two reasons: 1) To reduce the computational cost of a fully connected neural network. 2) To reduce overfitting in the network.

Given a large number of post-training pruning approaches Srivastava et al. (2014), Geman et al. (1992), Pan et al. (2016), the paper attempts to propose two pre-training pruning approaches: kstarts and dissipating gradients. Moreover, it appears to be the case that when isolated from other factors sparse networks outperform fully connected networks. When not isolated they perform at least as well up to a percentage of sparsity depending on the number of parameters in the said network. kstarts and dissipating gradients provide are simple nevertheless effective methods to quickly look for best sparse networks to.

The approaches exploit the knowledge that a network has multiple underlying p-sparse networks that perform just as well and in some cases even better when contrasted with their fully connected

counterparts Frankle & Carbin (2018). What percentage of sparsity is realized, depends largely on the number of parameters originally present in the network. Such sparse networks are potent in preventing over-fitting and reducing computational cost.

The poset-training pruning has several approaches in place such as adding various regularization schemes to prune the network Louizos et al. (2017), Pan et al. (2016) or using second derivative or hessian of the weights for dropout LeCun et al. (1990), Hassibi & Stork (1993). Han et al. (2015), Alford et al. (2019), Zhu & Gupta (2017) use an efficient iterative pruning method to iteratively increase sparsity. Srivastava et al. (2014) dropout random hidden units with p probability instead of weights to avoid overfitting in general. Each of these approaches is effective and achieves good sparsity post-training.

We use a simple intuitive models that achieve good results and exploits the fact that a number of sub networks in a Neural Network has the potential to individually learn the input Srivastava et al. (2014). We decide on a sparse network early on based on the dropout method and use only that for training. This provides an edge for faster computation, quicker elimination of excess weights and reduced generalization error. The sparsity achieved is superior to random dropout.

Section II gives a general introduction to all the methods, section III defines p-sparsity, section IV provides the algorithm for both approaches, section V describes experimental setup and results, section VI discusses various design choices, section VII gives a general discussion of results, section VIII discusses limitations of the approach and section IX provides conclusions and final remarks.

## 2 PRUNING METHODS

### 2.1 KSTARTS

#### 2.1.1 KSTARTS AND EVOLUTIONARY ALGORITHMS

We take the concept of k random starts from Evolutionary Algorithms (Vikhar, 2016) that use a fitness function or heuristic to perform "natural selection" in optimization and search based problems (Goldberg & Holland, 1988). It is relatively simple to fit genetic algorithms to the problem at hand. Other method that would be equally effective with a little bit of modification are Hunting Search (Oftadeh et al., 2010), Natural Adaptation Strategies (Wierstra et al., 2008), firefly algorithm (Yang, 2010) etc.

The basic components of the algorithm are: (1) **Population**: A product of network weights and sparse matrices. (2) **Individual**: An instance of the population. (3) **Fitness Function**: The heuristic chosen for evaluation of the population.

#### 2.1.2 POPULATION

We first initialize K sparse matrices, a single instance of these K sparse matrices can be seen in equation **??**. In every iteration we multiply model weights W of the Network layer in question with every instance of the K sparse matrices. The resulting set of matrices is our population for that iteration. Each iteration is referred to as a new generation.

$$population = W * K - SparseMatrices \qquad (1)$$

#### 2.1.3 INDIVIDUAL

Each individual, in a population of K instances, is a sparse matrix of size equal to the size of network weights, W. The number of 0's and 1's in the sparsity matrix are determined by the connectivity factor p which is further described in section 3. An sparse matrix of $p \approx 0.5$ will have $\approx 50\%$ 0's and $\approx 50\%$ 1s.

### 2.1.4 EVALUATION/FITNESS FUNCTION

The fitness of an individual is ranked by determining the sum of each individual in population as given in 1 such that the fittest individual in a generation is given by equation 2.

$$fittest = \arg\max_{ind} \sum_{j=1}^{i*c} ind[j](\text{ of population}) \tag{2}$$

where $i * c$ is the size of each individual and ind refers to the individual in population.

### 2.1.5 NEXT GENERATION SELECTION

Assuming each iteration is the next generation. In each generation the fit individual is favoured so:

- The fittest individual is passed of as weight to the next generation.
- Every 5 generations or as per the decided elimination frequency, the individual with lowest fitness is discarded from the population.

### 2.2 DISSIPATING GRADIENTS

Magnitude and gradient based pruning approaches are popular in post-training pruning Srivastava et al. (2014), LeCun et al. (1990). It does not make much sense to employ them pre-training becauseof the randomly generated weights. But in order to reduce error, any network aims to target updating weights that influence the results most. Based upon that hypothesis, in each epoch we sum gradients over and eliminate weights whose weights are not getting updated. In equation 3 N is the total number of iterations for an epoch. In equation 4 epsilon is 1e-6 for all experiments.

$$Accumulated\_dw = \sum_{i=1}^{N} dW \tag{3}$$

$$W[Accumulated\_dw < \epsilon] = 0 \tag{4}$$

One consideration in this approach is to not do this for too many epochs which can be only 2 if the image is very monochrome and more than 2 if the gradients are dissipating more slowly. Moreover, once specific weights have reached their optimal learning, their gradients will dissipate and we don't want to eliminate them.

### 2.3 COMBINATION DROPOUT

Combination dropout is merely combining Kstarts with dissipating gradients. The weights eliminated use both approaches. We fix p for Kstarts to a certain value of minimum sparsity and further eliminate weights that dissipating gradients method will eliminate as well. The approach achieves better performance than either methods.

## 3 DEFINING P-SPARSITY

In a p-sparse network layer approximately p percent of connection between two layers are eliminated. Figure 1 shows a fully connected conventional neural network (figure 1a) and three sparsely connected networks with different values of p (figures 1b, 1c, 1d).

### 3.1 CONNECTIVITY FACTOR

Connectivity factor, p, determines the percentage of connections to be removed between any two layers of the network. For instance if p = 0.0 than the network is be fully connected as shown in figure 1a, if on the opposite extreme, p=1.0, then there will be no connections between two layers of neurons. If p=0.5, figure 1b, only approximately 50% of the connection in the network remain. If p=0.3, figure 1c, approximately 70% of the connection still exist and if p=0.7, figure 1d, a mere 30% of the connections are active between two network layers. In short p determines percentage of 0's in an individual sparse matrix shown in equation **??**.

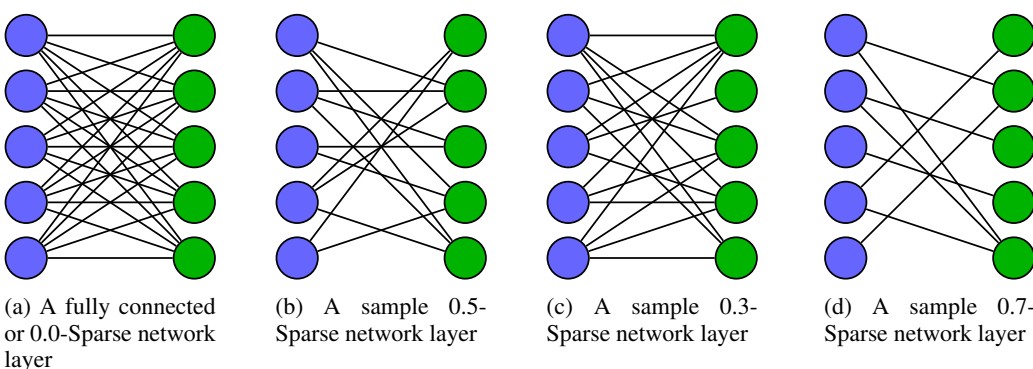

(a) A fully connected or 0.0-Sparse network layer

(b) A sample 0.5-Sparse network layer

(c) A sample 0.3-Sparse network layer

(d) A sample 0.7-Sparse network layer

Figure 1: Sample P-Sparse Network Layers

## 4 ALGORITHM

The autoencoder, two and three layered neural networks are trained in a standard manner with adam optimizer and batch update.

### 4.1 KSTARTS ALGORITHM

Algorithm 1 gives the kstarts method:

1. We have K number of sparse matrices as shown in equation **??** generated and we call them KI in algorithm 1.

2. Every time the weight W needs to be updated instead of the usual gradient update we add one step further and using the fitness function, pass the fittest individual as the new W which biases the network towards that sparse matrix.

3. Every 5 iterations, the individual with lowest fitness is dropped.

---

**Algorithm 1:** k Random starts

**input** : Data, params,K,p
**output:** W,b
initialize W,b;
$KI \leftarrow$ K individuals with approximately 1-p percent active connections;
**for** *maxiterations* **do**
    Run Neural Network;
    Update weights;
    **if** *One individual in KI is left* **then**
        $W \leftarrow$ individual;
    **else**
        $W \leftarrow$ individual with maximum sum (of weights);
        **for** *every 5 iterations* **do**
            pop individual with minimum sum (of weights) from KI;
        **end**
    **end**
**end**

---

### 4.2 DISSIPATING GRADIENTS ALGORITHM

Algorithm 2 is more simple and just eliminates weights with sum of gradients equal to zero in the first 1-4 epochs depending on the desired sparsity.

---

**Algorithm 2:** Dissipating Gradients

---

**input** : Data, params
**output:** W,b
initialize W,b;
**for** *maxepochs* **do**
    **for** *Maxiterations* **do**
        Run Neural Network;
        accumulated_dW ← accumulated_dW+ dW;
        Update weights;
    **end**
    **if** *accumulated_dW* $< 0.0001$ **then**
        accumulated_dW←0 ;
    **else**
        accumulated_dW ←1;
    **end**
    W ← W*accumulated_dW;
**end**

---

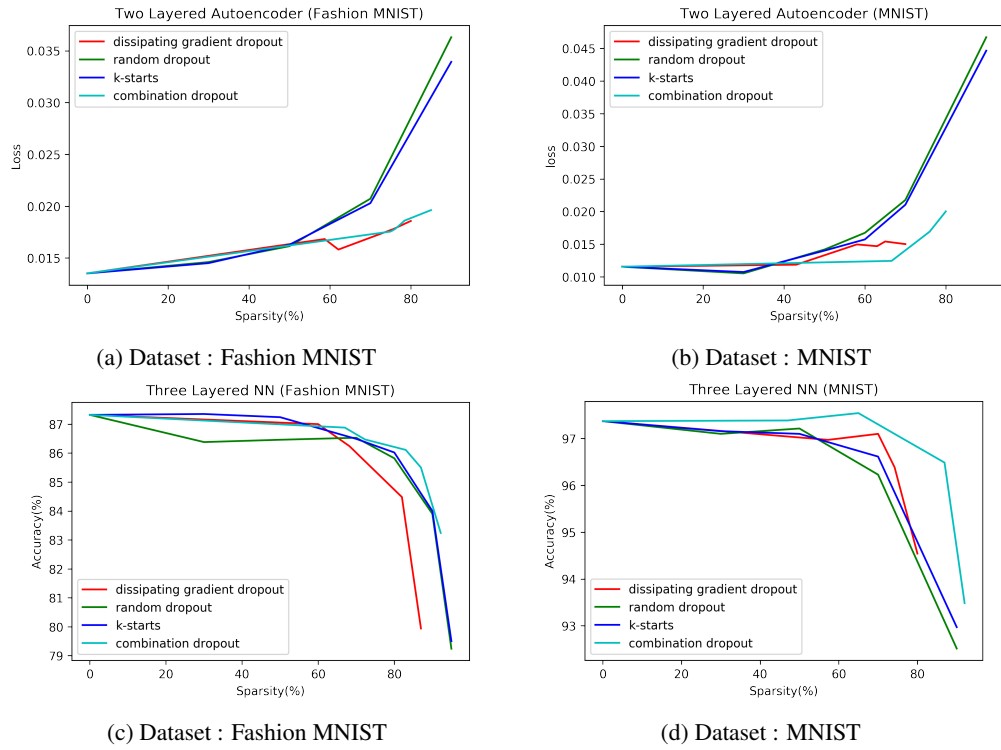

(a) Dataset : Fashion MNIST          (b) Dataset : MNIST

(c) Dataset : Fashion MNIST          (d) Dataset : MNIST

Figure 2: Performance of dropout methods with increasing sparsity.

# 5 EXPERIMENTS AND RESULTS

The experiments performed on two datasets; MNIST Deng (2012) and Fashion MNIST Xiao et al. (2017). The network architectures are two layered Autoencoder (784-128-64), a three layered NN (784-128-100-10) both with sigmoid activation and adam optimization functions.

The Architecture used for learning curves is a single layered NN(784-10).

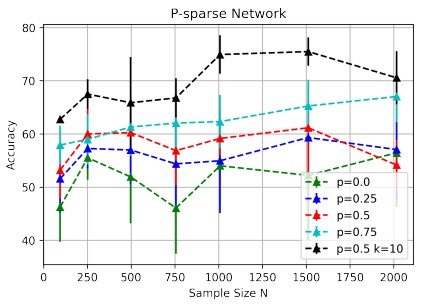
(a) Dataset : Fashion MNIST

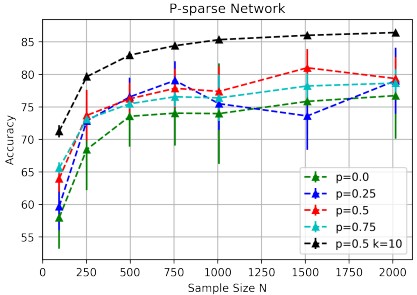
(b) Dataset : MNIST

Figure 3: Effect of sample size on learning when varying the sparsity of the network with kstarts. Each trial involves running a p-sparse network on a fixed training set of N examples from dataset for 2.5k iterations and is averaged over 10 trials each.

## 5.1 EFFECT OF INCREASING SPARSITY

As sparsity increases, overall performance reduces. Figure 2 shows the behaviour of various dropout methods presented in this paper.

In case of random dropout, it's indeed a random shot. Either no useful weight is eliminated or multiple crucial weights are eliminated which decides how well does random dropout perform. Kstarts performs slightly better on average with multiple start choices. Depending on how many independent p-sparse networks are present in the network that can learn well, one of them can be identified, given k is large enough and the fitness function is smartly decided by first examining the weights and gradients.

Dissipating gradients works well as long as the network isn't learning very fast i.e. some weights are being updated in the consequent epochs. It's also most reliable. Combination works by far the best because it does not only rely on eliminating weights that are not being updated but also uses kstarts. It seems to achieve superior performance as long as p value chosen is a value that kstarts performs well on.

## 5.2 VARIATION IN SAMPLE SIZE

Figures 3a and 3b show relationship of varying sample size to different sparsity values in a single layer NN over 2.5k iterations.

The interesting result here is that isolated from all other factors like number of parameters, hidden units and various design choices, kstarts dropout performs better on a single layer network compared to even a fully connected network. The standard deviation is also lower to fully connected network as well as random dropout. kstarts dropout also learns faster than a fully connected network. For instance, if the iterations for the experiments in figure 3a and 3b are increased, the fully connected network will eventually reach accuracy of the p-sparse network.

## 6 DESIGN PARAMETER CHOICES

There are a number of design parameter choices for the algorithm 1 presented here. Some are explained in detail in this section.

## 6.1 CHOICE OF FITNESS FUNCTION

Since the fitness function determines the individual being passed on to the next generation and the individual being eliminated. We had three choices for choosing the fitness of an individual each with it's own pros and cons.

- **Magnitude:** As opted here, we choose population as shown in equation 1 and then select fitness using equation 2. This skews the selection of new weights to the previously selected sparse matrix from KI and therefore, the initial sparse matrix will be propagated forward. This also renders elimination to be pointless it does not matter if 1 or all other matrices in K are eliminated. Furthermore, the sparse matrix is picked awfully early in the experiments i.e. only after first 5 iterations or so and that is not when weights have reached a saturation point.

- **Gradient:** The second choice is to use gradient of the weights to create population as follows:

$$population = \delta W * SparseMatrices \tag{5}$$

  doing so can have fitness totally dependant on the current update and the new weights are heavily skewed towards the performance of the current iterations which again doesn't seem that appropriate.

- **Sum of Gradients:** The third option is summing up the gradients for a number of iterations in the manner we do for dissipating gradients 3 and then use those to create population:

$$population = \left(\sum \delta W\right) * SparseMatrices \tag{6}$$

  Doing so skews the network toward weights that are updated quickly and are increasing.

## 6.2 No. of layers and no. hidden units

We initially use single layer NN (784-10) to isolate the effects of kstarts algorithm from effects of other parameters that may have a large impact on performance as the size of the network grows. Those parameters i.e. number of hidden layers, regularization choices, types of layers may aid or adversely effect the performance of the algorithm and by performing the experiments on the basic unit of a neural network we were able to concur that the method is effective in it's own right. After concluding the approach works we tested on three layered NN and Autoencoder.

## 6.3 Cost Effectivesness

One beneficial feature of pre-training pruning approaches is that the best p-sparse network is quickly identified. There are a number of methods that exploit sparsity of a matrix for quicker multiplication i.e. (Yuster & Zwick, 2005), (Buluç & Gilbert, 2012) which can be used to quickly retrain large networks. Although that is out of scope for our findings.

## 7 Discussion

### 7.1 Effect of K

1. From the experiments done so far lower K ($\approx 10$) outperforms or at least performs as well as a higher value of K ($\approx 100$). This can be because the more times a different matrix is chosen by the network, the more times the network has to adopt to learning with only that p-sparse network.

### 7.2 Effect of P

1. A sparse network can outperforms a fully connected network for lower number of iterations and smaller networks.

2. An appropriate value of p, for instance in these experiments $p \approx 0.5$, seems to work best for random dropout, kstarts dropout and combination dropout. A poor choice of p can't seem to be remedied by a better choice of K.

3. p can be thought of as an information limiter. For better learning, if the network is only provided with particular features, it might have an easier time learning the class specific features in different nodes but this only remains a wishful speculation and requires further analysis. Table 1 shows relationship between k,p and no of iterations in a single layer NN (784-10)

Table 1: MNIST-Mean Accuracy and standard deviation averaged over 10 runs

| | p=0.0 | p=0.5 | | | |
|---|---|---|---|---|---|
| | | k=1 | k=10 | k=50 | k=100 |
| iter 10 | $12.29 \pm 3.79$ | $13.36 \pm 4.17$ | $13.94 \pm 3.52$ | $9.71 \pm 4.04$ | $10.14 \pm 0.63$ |
| iter 100 | $28.41 \pm 7.42$ | $28.96 \pm 3.03$ | $50.53 \pm 10.7$ | $46.19 \pm 10.10$ | $14.83 \pm 6.169$ |
| iter 1k | $60.71 \pm 8.77$ | $63.39 \pm 6.94$ | $82.11 \pm 2.97$ | $81.77 \pm 4.27$ | $82.13 \pm 3.63$ |
| iter 10k | $82.49 \pm 4.3$ | $88.40 \pm 0.51$ | $89.07 \pm 0.4$ | $89.08 \pm 0.38$ | $89.21 \pm 0.38$ |

## 8 LIMITATIONS

There are a number of limitations to the approach and further investigation is required in a number of domains.

1. The NNs used are single and three layered feed forward networks and autoencoders. CNNs are not experimented upon.

2. Only classification tasks are considered and those on only two datasets: MNIST and Fashion MNIST.

3. Ideally using sparse matrix should make for efficient computation but since the algorithms for that are not used it at this point does not show what the time comparison of the approaches will be.

## 9 CONCLUSIONS AND FINAL REMARKS

We present two methods for pruning weights pre-training or in the first couple of epochs. The comparisons are made against random dropout and both approaches mostly perform better than the random dropout. We provide a combination dropout approach that consistently outperforms other dropout approaches. A lot more analysis of the approach is required on multiple datasets, learning tasks and network architectures but the basic methods seem to be effective.

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
