# OpenReview forum: "What to Prune and What Not to Prune at Initialization"
_ICLR.cc/2021/Conference — Reject_

### Official Review · AnonReviewer3 · 2020-10-14
**I think this paper still needs more efforts, in terms of both experiments and paper writing. I give a clear rejection.**

**Rating:** 3
**Confidence:** 5

**Review:**

Overview:

This paper proposed two methods, K-starts and dissipating gradients approach to sparsity the network before training and achieve better performance than a random dropout.


Strength bullets:
1. The idea is interesting, but needs more effort to complete it.


Weakness bullets:
1. poor writing, for example, citation error in abstract line 3 //? exist in 2.1.2 line 2 and the last line of 3.1  // the explanation of sparsity is colloquial in 2.1.3
2. As for SparseMatrices in equation 1, does it mean the explanation of K? It's easy to confuse with a minus sign.
3. because the classification on MNIST and FashionMNIST is too simple, the improvement of k-starts is marginal
4. As dissipating gradient dropout needs training for a couple of epochs, it's unfair to compare with random dropout.
5. The fatal limitation is the lack of comparison with previous methods and related works, need to compare with other pruning from scratch methods, like GraSP (https://arxiv.org/abs/2002.07376), SNIP (https://arxiv.org/abs/1810.02340).

---

### Official Review · AnonReviewer4 · 2020-10-29
**Novel pruning techniques but limited in scope**

**Rating:** 4
**Confidence:** 4

**Review:**

Summary: This paper looks at two new pruning methods, k-starts and dissipating gradients, as well as a combination of both. They both perform better than random, and the combination performs the best.

Pros
* K-starts and dissipating gradients are novel techniques to my knowledge.
* It makes sense that weights with zero gradient over time can be considered unimportant (though this could mean that you should freeze the weight at its current value rather than prune it and set it to zero). It is also a promising idea to prune early based on gradient info from the early epochs.

Cons
* The results of this paper are limited to MLPs MNIST and Fashion MNIST. Methods that work on these small datasets do not always generalize well to other datasets. MNIST with MLPs is a particularly special case for pruning because of input pixels that are always zero. I would expect to see at least CIFAR-10 and some convolutional networks.
* While the techniques are compared to each other and to random pruning, there are no comparisons to other existing pruning methods.
* Some unclear points in writing: pruning seems to be conflated with “dropout”, even though they are two very different techniques. The authors say they use autoencoders, but I believe they should be MLPs if the task is MNIST.
* Figure 2 should have error bars. Is this test or train loss/accuracy?
* Section 7.2: performance at different sparsity levels has been studied by many other papers (e.g. https://arxiv.org/pdf/2003.03033.pdf), and it’s not a novel finding that accuracy can increase for some p, but large p causes performance to degrade.

Additional clarification questions:
* 2.1.2: text description does not match equation 1, do you multiply the model weights with K and then subtract a matrix as in the equation, or multiply model weights with each of the K matrices as in the text?
* What is the rationale behind the fitness function in 2.1.4? If you are optimizing for the sum of the individual’s weights (assuming that’s what ind[j] means), why not just optimize for that directly, without using multiple matrices and the evolutionary algorithm? Likewise, in algorithm 1, why not just prune by weight values directly?
* Equation 3: it supposed to be a sum over the absolute values of the gradients? If not, weights with strong negative gradients should be treated the same as those with strong positive gradients.
* Section 4: what are the details of “standard manner” and what is “batch update”?

Minor: there are some latex errors, such as “cite” in the abstract, invalid equation references, and incorrect quote formatting. The abstract should probably be shorter.

Overall: this paper poses two novel techniques for pruning, but do not evaluate them thoroughly on enough models and datasets and with comparisons to other pruning methods. Due to the limited scope, I do not recommend acceptance.

---

### Official Review · AnonReviewer2 · 2020-10-29
**A sketch of suggestions with no serious intention**

**Rating:** 1
**Confidence:** 4

**Review:**

Summary: The authors propose two approaches for pruning: (a) "Evolution-style": start with K random masks associated with the weights, update weights on gradient descent corresponding to those active in the “fittest” mask, and overtime throw away all but one masks which are less fit. (b) "Dissipating-gradients”: Here those weights are removed which are not being updated as much, measured by their sum of gradients over a number of iterations. This is shown for elementary networks on MNIST datasets without any serious experiments or comparisons or even presentation.


Pros:
-It is quite possible that these nuggets of “ideas” proposed in this paper could be useful, in some shape. But right now their form is *as rough as a draft can possibly get*.
-Another thing which is mentioned rightfully in their paper is the “Limitations” section, of which are plenty.

Cons:
Empirical analysis
-No serious experiments or comparisons to any baseline apart from random pruning. Even so, all networks used comprise of a few layers on MNIST & Fashion MNIST.
-The performance of their proposed methods is basically on par with random pruning, sometimes even worse (Figure 2)!
-Why do you sum over gradients? Doesn’t it make more sense to sum the absolute value of gradients? (There could be a weight which is changing a lot, but at the time when you decided to prune, its sum of updates was very small.)
-What happens in Figure 3a and 3b if you use all the samples from MNIST?
-Given the extremely rough shape of the paper, I don’t think if the other results should even be taken seriously!

Presentation:
-The authors keep saying post-training “dropout”, instead of post-training “pruning”. Dropout is just used as a verb in place of pruning. It’s not that authors do not know that Dropout and they cite the corresponding paper. Yet, it is used by the authors as if dropout and pruning are the same things.
-The equations are so poorly written, why even bother?
-The writing is absolutely dismal, I don’t even think I could enumerate over all the issues here.
-There are several vague statements: "It is also averse to retraining losses.”, "which can be only 2 if the image is very monochrome”, etc.
-There is one paragraph on the connectivity factor, which is trivial. It’s basically what fraction of weights are pruned. Funnily, this concept is accompanied with diagrams.


Sorry if this whole review sounds rude, but to be honest I am amazed that authors thought about submitting a paper in this state to ICLR.

---

### Official Review · AnonReviewer1 · 2020-11-02
**interesting ideas, but the paper is incomplete and the experiments are insufficient**

**Rating:** 2
**Confidence:** 5

**Review:**

# No Rebuttal

Since there is no rebuttal, I have not modified my score.

# Overall

This paper presents some interesting ideas. In particular, this is the first time I have seen genetic algorithms used in the context of pruning, and I encourage the authors to explore this concept further. However, the paper has some severe weaknesses:

(1) The paper doesn't appear to be complete. The latex is a mess, with broken references, missing formulas, citations that aren't in parentheticals, math notation that is broken, etc. Key aspects of the methodology (as mentioned in the notes below) are missing. This paper was not in submittable condition, and the authors should probably have continued to work on the paper and submitted it to another conference.

(2) The two methods studied in the paper have little to do with each other. It's unclear why they're both included in the same paper, and they should be studied separately. The dissipated gradients method has little justification itself, and it is simply declared.

(3) The paper only studies MNIST and FashionMNIST, which are known to allow for many findings that do not scale to larger networks. I encourage the authors to extend their method to a larger-scale setting (e.g., ResNet-20 on CIFAR-10) to show that the results also work there.

(4) The paper does not include any baselines, including other early pruning methods, making it impossible to evaluate whether these results are trivial or SOTA in any way. The authors cannot plead ignorance on this matter: they cite a paper (Frankle et al., 2020) in their abstract that explicitly surveys methods for pruning at initialization. In fact, one of the primary findings of Frankle et al. 2020 is that it's important to include baselines.

I cannot recommend this paper for acceptance, and I suggest that the authors re-submit the paper at a time when it is in more complete shape.

# Notes

## Abstract

The citations in the abstract are garbled.

The abstract mentions two approaches but three are given.

MNIST and FashionMNIST are not sufficient datasets to justify any pruning results. Many pruning results (e.g., lottery tickets) work on MNIST but not on larger-scale settings. The authors will need to include a larger-scale setting (e.g., CIFAR-10 and ResNet-20) to substantiate any results in this paper.

## Intro

The authors should use the "\citep" macro to ensure that citations are parenthetical rather than appearing in the middle of the text.

Pruning was generally claimed as a way to reduce overfitting in the 1980s and early 1990s, but it generally has not been used in this way in almost 30 years. Today, it is used solely to reduce costs (e.g., storage, running time, FLOPs, etc.).

Pruning is used on all sorts of networks, not just fully-connected networks.

The citations and sentence structure are garbled throughout this section.

## Section 2

I have no idea what Equation 2 means, but I assume you're keeping population members that lead to the highest accuracy before any training occurs?

I assume that, once an individual makes it to the next generation, its weights are perturbed in some way to create new members of that generation? This doesn't seem to be explained in the text.

What is the justification for dissipated gradients? Why this design and not other designs.

Dissipated gradients and the genetic algorithm (GA) approach are completely unrelated. Why are they in the same paper? Perhaps each should receive consideration in a separate paper.

Why combine these two approaches and not any two other approaches?

## Section 5

The paper frequently uses the word "Dropout" when it means "Pruning." Dropout is something different entirely.

## Section 6:

What does the magnitude heuristic mean? Why don't these fitness heuristics appear in Section 2?

---

### Decision · Program_Chairs · 2021-01-07
**Final Decision**

**Decision:**

Reject

**Comment:**

This work appears to be a promising start to a research direction. However, as the reviewers noted, the work does not compare to alternative approaches and the presentation of the work overall is incomplete.